# Anatomical Prior-Based Automatic Segmentation for Cardiac Substructures from Computed Tomography Images

**DOI:** 10.3390/bioengineering10111267

**Published:** 2023-10-31

**Authors:** Xuefang Wang, Xinyi Li, Ruxu Du, Yong Zhong, Yao Lu, Ting Song

**Affiliations:** 1Shien-Ming Wu School of Intelligent Engineering, South China University of Technology, Guangzhou 511400, China; 202010109503@mail.scut.edu.cn; 2Department of Radiology, The Third Affiliated Hospital of Guangzhou Medical University, Guangzhou 510150, China; lixinyilgd@stu.gzhmu.edu.cn; 3Guangzhou Janus Biotechnology Co., Ltd., Guangzhou 511400, China; duruxu@scut.edu.cn; 4School of Computer Science and Engineering, Sun Yat-sen University, Guangzhou 510275, China; 5Guangdong Province Key Laboratory of Computational Science, Sun Yat-sen University, Guangzhou 510275, China; 6State Key Laboratory of Oncology in South China, Guangzhou 510060, China

**Keywords:** CT, cardiac substructure segmentation, deep learning, medical image segmentation, anatomical knowledge

## Abstract

Cardiac substructure segmentation is a prerequisite for cardiac diagnosis and treatment, providing a basis for accurate calculation, modeling, and analysis of the entire cardiac structure. CT (computed tomography) imaging can be used for a noninvasive qualitative and quantitative evaluation of the cardiac anatomy and function. Cardiac substructures have diverse grayscales, fuzzy boundaries, irregular shapes, and variable locations. We designed a deep learning-based framework to improve the accuracy of the automatic segmentation of cardiac substructures. This framework integrates cardiac anatomical knowledge; it uses prior knowledge of the location, shape, and scale of cardiac substructures and separately processes the structures of different scales. Through two successive segmentation steps with a coarse-to-fine cascaded network, the more easily segmented substructures were coarsely segmented first; then, the more difficult substructures were finely segmented. The coarse segmentation result was used as prior information and combined with the original image as the input for the model. Anatomical knowledge of the large-scale substructures was embedded into the fine segmentation network to guide and train the small-scale substructures, achieving efficient and accurate segmentation of ten cardiac substructures. Sixty cardiac CT images and ten substructures manually delineated by experienced radiologists were retrospectively collected; the model was evaluated using the DSC (Dice similarity coefficient), Recall, Precision, and the Hausdorff distance. Compared with current mainstream segmentation models, our approach demonstrated significantly higher segmentation accuracy, with accurate segmentation of ten substructures of different shapes and sizes, indicating that the segmentation framework fused with prior anatomical knowledge has superior segmentation performance and can better segment small targets in multi-target segmentation tasks.

## 1. Introduction

According to a report from the World Health Organization (WHO), heart disease has been the leading cause of death worldwide for the past 20 years [1]. To effectively prevent and treat various heart diseases, accurate calculation, modeling, and analysis of the entire heart structure are essential for research and applications in the medical field. CT imaging, the current gold standard for the diagnosis of many diseases, is widely used in early diagnosis and screening of the brain and heart [2]. Using CT, it is possible to qualitatively and quantitatively assess the anatomical structure and function of the heart and provide support for diagnosis, disease monitoring, treatment planning, and recovery prediction. Cardiac image segmentation is the first step in these applications. It divides an image into multiple anatomically meaningful regions, from which quantitative metrics can be extracted [3]. Typically, the anatomical structures of interest in cardiac image segmentation include ten substructures: the left atrium (LA), right atrium (RA), left ventricle (LV), right ventricle (RV), superior vena cava (SVC), inferior vena cava (IVC), pulmonary artery (PA), pulmonary vein (PV), ascending aorta (AA), and descending aorta (DA). The circulatory system consists of the heart, arteries, and veins, which deliver blood throughout the body and are divided into pulmonary and systemic circulation. Cardiac substructures are the main forces of pulmonary and systemic circulation and are the basic elements for analyzing the entire heart structure and function.

Currently, segmentation of cardiac substructures in clinical practice relies heavily on manual labor. Radiologists manually draw contour lines of substructures, including the heart ventricles, atria, and vasculature, to obtain clinical parameters for analyzing heart function, such as ejection fraction and stroke volume. However, manual segmentation is often time-consuming and prone to subjective errors, resulting in a heavy workload for medical professionals and inconsistent segmentation results influenced by factors such as diagnostic experience. With a gradual increase in the incidence of cardiac-related diseases, manual segmentation cannot meet the great demand for medical treatment. A fast and accurate, fully automated cardiac segmentation algorithm is necessary to improve diagnostic efficiency, particularly in clinical applications of cardiovascular surgery planning and radiation therapy.

The early studies on automatic cardiac substructure segmentation were based on traditional machine learning methods and mainly relied on visible image features such as image texture, image edges, and pixel intensity to achieve segmentation of the heart target [4,5,6,7,8,9,10,11]. Hautvast et al. designed a new active contour model for segmentation of cardiac ventricles. This segmentation method obtains a constant contour by comparing it with the grayscale value of the vertical contour. The contour maintains a relatively constant position with the surrounding tissue to achieve better segmentation results [9]. Billet et al. used coupled and active models to restrict the motion of the endocardium and epicardium to achieve biventricular segmentation of the heart. This method uses special image features, image intensity, and gradient information to achieve better cardiac motion recovery and higher segmentation accuracy [10]. Senegas et al. proposed a complete Bayesian model that represented prior information regarding the shape and motion of the heart. The superiority of this method for segmentation was demonstrated by the probability distribution of the parameters and the results of Monte Carlo methods [11]. These segmentation methods are susceptible to the influence of similar feature tissues around the segmented heart target, due to factors such as noise. Traditional segmentation models rely on large amounts of prior knowledge. Modeling usually assumes that the heart is stationary and that the shape of the ventricle is cylindrical and requires manual feature design and segmentation. However, in actual conditions, different parts of the heart are constantly in motion; this leads to the weak performance generalization and poor robustness of the traditional models. The traditional methods also have a common insurmountable problem: feature extraction must be designed and performed manually; thus, it is highly dependent on the domain-specific knowledge of the researchers. This process produces a significant workload and greatly affects the quality of the features, which can lead to significant errors in the segmentation results. Most traditional methods are semi-automated methods that require manual interaction and cannot simultaneously segment three different cardiac structures; they can only customize them for a cardiac region. Thus, these methods do not satisfy the requirements for real-time processing.

In recent years, deep learning has been widely used in the field of medical image segmentation owing to its advantages in automatic feature extraction and nonlinear modeling. Its segmentation accuracy exceeds that of the traditional segmentation methods [12]. By integrating new technologies, deep learning achieves higher segmentation accuracy by learning from limited datasets, with fully automated batch processing of images, which greatly relieves the pressure on physicians in the interpretation of medical images. Many studies have achieved good results in the segmentation of the heart, especially in the segmentation of heart chambers (such as the atria and ventricles LV, RV, LA, and RV); many excellent algorithms have emerged [13,14,15,16,17,18,19,20]; however, they are mostly for one or a few substructures. Some studies have been based on the FCN [21], which uses convolutional layers to replace the last fully connected layer of the CNN to achieve pixel-level classification and end-to-end semantic segmentation of images. However, the pooling operation of the FCN causes loss of pixel position information, resulting in poor segmentation details. Olaf et al. proposed a U-Net based on an encoder–decoder structure, using skip connections between the corresponding layers of the encoder and decoder networks to help the decoder network to better recover the spatial information of the target pixels and achieve reasonable segmentation accuracy with fewer training iterations [22]. Its performance exceeds that of an FCN; a series of encoder–decoder networks derived from U-Net have subsequently been proposed and successfully applied to heart segmentation. In Reference [23], a dense version of U-Net with an initial module was developed to combine multiscale features for robust segmentation of images with large anatomical variations. Zotti et al. proposed an optimized U-Net model that embedded prior information on the shape of the heart to achieve a fully automated MRI heart image segmentation method [24]. To further improve the segmentation performance of the heart, some studies have guided 2D networks by introducing information on different time ranges of the cardiac cycle [25]. Researchers have also proposed multistage network segmentation methods that decompose the segmentation problem into subtasks to first locate the ROI (region of interest) and then connect the segmentation network to refine the segmentation [26]. These studies focused on segmentation of the atria and ventricles and can be used to evaluate their function and examine their volumetric changes, with great clinical significance for early diagnosis of heart disease. However, for computer-aided diagnosis, treatment, and the actual needs of physicians, it is often necessary to segment the entire heart.

With the emergence of the multimodality whole-heart segmentation (MM-WHS) challenge, an increasing number of methods have begun to focus on the automatic segmentation of cardiac substructures, benefiting from the availability of public datasets for whole-heart segmentation (including the four chambers, aorta, and pulmonary veins). Payer et al. [13] used a two-step segmentation process; one step located the heart position and the other narrowed the segmentation range for finer segmentation. Wang and Smedby [27] proposed a fully automatic algorithm for whole-heart segmentation by incorporating prior knowledge of the shapes of the heart substructures into a jointly trained convolutional neural network, achieving a good performance. Yang et al. [28] used a generic framework based on three-dimensional fully convolutional networks, focusing their work on model training and achieving good results. There are several problems with deep learning-based cardiac CT image segmentation. First, many previous studies used neural network models in cardiac substructure segmentation without considering cardiac anatomical structural information. Second, other segmentation methods introduce a small amount of prior information to enhance model performance by embedding additional modules into the neural network in the case of limited data. Through these novel designs, the performance of the model has been improved to some extent, but these methods lack effective anatomical significance from a mathematical perspective, such as using the perimeter of the target or the relationship between adjacent points as regularization to restrict the optimization process. However, some prior knowledge is difficult to represent using regularization [29]. Some methods represent a shape using parameters (such as the center of gravity and radius) or non-parametric methods (such as level set representation), which can perform better in segmenting organs with relatively fixed shapes. However, the shape of the same organ in different individuals may differ and cannot be described using the same parameters [30,31]. Some methods describe the shape using distributions (such as Gaussian and Gaussian mixtures), which represent the possibility of deformation on the main shape to some extent and partially solve the problem of differences in the shapes of the same organ in different individuals, but the statistical distribution may contain errors. In medical images, the basic structure of the body is the same, and the positions and shapes of the organs are generally fixed. These uniform image features are known as the anatomical priors of medical images and are helpful in analysis. Anatomical priors can handle segmentation errors caused by image noise and improve the algorithm reliability. Most methods are manually designed or use artificial prior information that lacks explanation and ignores the inherent anatomical prior structure of the image. However, this knowledge is helpful in improving network performance; suitable prior knowledge must be selected according to the algorithm. It is preferable to obtain prior knowledge through learning. Third, existing deep learning methods still have a common problem. They typically treat large and small substructures equally and rely on high-performance hardware platforms and large amounts of labeled data for model training. This leads to poor segmentation accuracy for small substructures with low contrast. Some methods assign higher weights to small organs in the loss function; however, these methods are not optimized for imbalanced organ segmentation and cannot solve the problem thoroughly.

In addition, inherent problems in cardiac CT images pose significant challenges to segmentation. There are many substructures in the heart; the shape and structure are complex, with large differences in size among the substructures. The substructures are close to each other, and the grayscale images of the structures are very similar, with weak contrast between them. Additionally, some substructures have blood inflow between them. Owing to differences in imaging principles and tissue characteristics, image formation is easily affected by factors such as noise and tissue movement. Moreover, the shape and position of the heart vary from person to person; even in the same person at different times or in different medical image slices, the shape and position of the heart may differ. Addressing these challenges by incorporating complex cardiac anatomical knowledge into the model to improve the segmentation accuracy without decreasing the model efficiency is a critical issue. We observed how physicians annotated the cardiac substructures. First, physicians delineate larger or more easily segmented substructures on a normal scale. For smaller substructures or those that are difficult to segment, physicians first determine the location of the small organ and then focus on the surrounding area to achieve a more accurate delineation, solving the problem of size differences and class imbalances. Physicians introduce prior anatomical knowledge of the heart structures during delineation to prevent annotation errors due to insufficient image information or imaging quality.

To address the limitations of the current deep learning segmentation models, our design, inspired by the approach of the doctors, incorporates prior anatomical knowledge of the position, shape, and scale of the cardiac substructure to separate different scales using a hierarchical segmentation approach to improve the segmentation performance of different scales, especially the sensitivity of small scales. We propose a novel framework for cardiac substructure segmentation that incorporates prior anatomical knowledge. The framework employs two consecutive segmentation steps using a cascade network from coarse to fine segmentation to separately address relatively easy and difficult-to-segment substructures. To improve the segmentation accuracy of small-scale substructures, we used the coarse segmentation results as prior information in combination with the original image to guide and train the fine segmentation network; we embedded anatomical knowledge of large-scale substructures into the fine segmentation network, which resulted in the efficient and accurate segmentation of ten cardiac substructures. The experimental results show that our proposed method has superior segmentation performance and can better segment small targets in multitarget segmentation tasks without requiring additional computation.

The remainder of this paper is organized as follows. Section 2 introduces the model structure and explains each component in detail. In Section 3, we describe the experiments, explore the effects of different grouping strategies on the segmentation results, and compare the results with those of the current mainstream models. Section 4 describes the typical problems encountered in this field and discusses the experimental results. Finally, we analyze the limitations of the experiment and provide prospects for future research.

## 2. Materials and Methods

### 2.1. Data Source and Data Analysis

With the approval of the Institutional Review Board, we collected cardiac CT data (DICOM) from 60 patients at the Third Affiliated Hospital of Guangzhou Medical University for this clinical study. Each case was annotated with ten structural contours (Figure 1), including the LV, RV, LA, RA, SVC, IVC, PA, PV, AA, and DA. The annotations were performed by two experienced radiologists, one mid-career doctor conducting the target contour annotation, and one senior doctor reviewing it. Statistical analysis of the annotations yielded the results shown in Figure 2. The proportion represents the average proportion of each substructure in each case; the average size refers to the average number of pixels occupied by each substructure in each slice of the case.

Several characteristics were identified from the experimental data analysis. First, the dataset is not large; the data are 3D, with a class imbalance, particularly for the smaller substructures. Second, there are targets of various sizes, with large differences between the largest (LV) and smallest (SVC) substructures. The left and right ventricles and atria were larger; the pulmonary veins and superior and inferior vena cava were smaller. Third, the cardiac substructures have diverse grayscale values, indistinct borders, irregular shapes, and non-fixed positions. Moreover, the background occupies the largest area in the CT images, which leads to significant imbalances between the substructures and between the substructures and the background, posing substantial challenges for neural network training for segmentation tasks.

### 2.2. Cardiac Substructures and Their Anatomical Information

The heart is located between the two lungs, near the middle of the chest, and slightly to the left. Its shape is similar to that of a peach and its size is roughly equivalent to that of a clenched right fist. A three-dimensional model and a sectional diagram of the entire cardiac anatomy are shown in Figure 3. The heart has four main chambers: the left ventricle, left atrium, right ventricle, and right atrium. In the human body, the left atrium and ventricle are on the left side; the right atrium and ventricle are on the right. The left and right sides of the heart are not connected and have similar structures, with upper empty spaces (atria) and lower spaces (ventricles). These four chambers comprise more than half the volume of the heart. The aorta is connected to the left ventricle; the pulmonary artery is connected to the right ventricle. The pulmonary veins connect to the left atrium; the superior and inferior vena cava connect to the right atrium. The right and left atria collect deoxygenated blood from the body and lungs, respectively. The right and left ventricles pump blood out of the heart for distribution to the body and lungs. Due to the interconnectedness of the cardiovascular system, CT can more clearly demonstrate the circulation of each component of the heart through continuous imaging. The anatomical information of the different substructures segmented in the cardiac images is presented in Table 1.

From Figure 3 and Table 1, we observe that the cardiac substructures have fixed relative positions and shapes. For example, in the four-chamber view of a transverse CT scan, the left ventricle is located in the upper left ventricle, the right ventricle in the upper right ventricle, the left atrium in the lower left ventricle, and the right atrium in the lower right ventricle. The left ventricle is located below the left atrium and to the left posterior of the right ventricle, whereas the right atrium is located to the right and front of the left atrium. The ascending aorta is connected to the left ventricle with a circular structure, and the pulmonary artery exhibits a tree structure. The inferior vena cava is typically located at the bottom of the heart. These insights provide the features of the various substructures of the heart; physicians can use these anatomical priors to annotate each structure with improved accuracy. We hope to design a segmentation model that integrates anatomical priors to improve the accuracy of the segmentation of cardiac substructures.

### 2.3. Anatomical Prior-Based Automatic Segmentation Framework

In this study, a cardiac substructure segmentation framework that integrates anatomical structure priors (Figure 4) was proposed to segment ten substructures in cardiac CT images. The framework consisted of two stages: coarse and fine segmentation. Larger or easier-to-segment substructures are segmented first to resolve the imbalance, before smaller or more challenging substructures are segmented. The coarse segmentation results are used as prior information; together with the original image, they form the input for the fine segmentation network. Anatomical knowledge, such as the location, shape, and scale of large-scale substructures, is embedded into a fine segmentation network through geometric constraints, helping to train small-scale substructures and improve the segmentation accuracy of small targets.

U-Net is particularly suitable for the semantic segmentation of datasets at a small scale due to its simple and symmetric structure. It can combine information from low- and high-level layers, fuse features of multiple scales, and achieve excellent segmentation results. Thus, our model uses U-Net as the backbone and divides the ten segmentation targets of the cardiac substructures into two groups, the coarse segmentation group (CG) and the fine segmentation group (FG), according to the characteristics of the cardiac substructures. These two segmentation subtasks are cascaded; the output of the coarse segmentation module serves as an input for the fine segmentation module. This cascading strategy facilitates training of each level of the network model. Using the results from the previous level of network segmentation to constrain the areas segmented by the subsequent level of the network can effectively reduce the number of false positives outside the defined area.

#### 2.3.1. Large Substructure Segmentation Network (LS-Net)

The coarse segmentation network LS-Net (purple area in Figure 5) segments the substructures in the CG and consists of mainly symmetrical contraction and expansion pathways. The contraction pathway obtains image context information through convolution and pooling structures, whereas the expansion pathway uses upsampling and convolution structures to obtain a segmentation target mask. Skip connections combine both low- and high-dimensional features. The contraction pathway contains four encoding modules, each consisting of two 3 × 3 convolutional layers with a rectified linear unit activation function, followed by maximum pooling to downsample the operation. Each downsampling reduces the size of the feature map by half. The expansion pathway is structurally similar to the contraction pathway and contains four decoding modules, each consisting of two ReLU-activated 3 × 3 convolutional layers. The feature map is upsampled to twice its original size. Skip connections between the contraction and expansion pathways combine features at different levels. The output module of the network includes two 3 × 3 convolutional layers; a dropout layer is added to prevent overfitting. Finally, a SoftMax-activated 1 × 1 convolution maps all the feature maps to the target class to obtain the final segmentation result.

Compared with the original U-Net, there were two improvements: (1) To avoid gradient diffusion during network training, batch normalization was added to all convolution layers except the last layer. (2) A dropout layer was added before the output layer to prevent overfitting.

#### 2.3.2. Small Substructure Segmentation Network (SS-Net)

The fine segmentation network SS-Net (blue area in Figure 5) segments the substructures in the FG group and consists of contraction and expansion pathways. The encoding and decoding modules are essentially the same as those of LS-Net; however, SS-Net cascades the segmentation results of LS-Net. We combined the mask output from LS-Net with the original slices and labels of the substructures in the FG group to form a multichannel input for network training. By encoding the segmentation results of large substructures, SS-Net embeds prior knowledge, such as the position, shape, and scale of large substructures. These high-resolution feature maps can help refine the segmentation results, particularly for small substructures.

We integrated the segmentation results of the small and large substructures to obtain the final prediction of all the substructures. This fully used the relatively stable anatomical information between substructures, reduced the difficulty of network training, and improved the segmentation accuracy.

#### 2.3.3. Grouping Strategy

For the division of the coarse segmentation group and fine segmentation group for the ten substructures of the heart, we designed three grouping methods (Table 2) and selected one of them as the optimal segmentation grouping strategy based on the segmentation results. The first grouping method was established based on the anatomy of the heart and the proximity relationship between the substructures. The left ventricle, right atrium, ascending aorta, descending aorta, superior vena cava, and inferior vena cava were grouped as the CG group. The left atrium, right ventricle, pulmonary artery, and pulmonary vein were grouped as the FG group. The rationale for grouping was that there are two blood circulation systems in the heart during surgery: systemic circulation and pulmonary circulation. The blood vessels that make up the systemic circulation include the aorta, which originates from the left ventricle of the heart, and the veins that return blood to the heart, such as the superior and inferior vena cava. The blood from the left ventricle is ejected into the aorta, and then travels along the arteries to the capillaries throughout the body. Subsequently, it drains into the small veins, large veins, and ultimately returns to the right atrium via the superior and inferior vena cava. The second grouping method was based on the size of the substructures. Given the heart cavity structure, the heart cavity was grouped into the CG group, which included the left atrium, right atrium, left ventricle, and right ventricle. The other blood vessels were grouped into the FG group, which included the superior and inferior vena cava, pulmonary artery, and aorta. The basis for grouping was that the relative positions of the four larger targets (left and right atria and ventricles) were fixed, whereas the arteries, veins, and other blood vessels were relatively small, connected, and closely related. The third grouping method grouped the left atrium, left ventricle, and descending aorta in the CG group; the right atrium, right ventricle, superior vena cava, inferior vena cava, and pulmonary artery were included in the FG group. This method explored whether grouping substructures with no connection to each other could help to improve segmentation performance.

### 2.4. Loss Function

Owing to the large difference in the proportion of targets and background in cardiac CT images and the large differences in size between the substructures, a hybrid loss function that combined the Dice loss [32] and SD loss [33] was used. The hybrid loss function has the optimization characteristics of both loss functions, namely optimization processing for class imbalance and sensitivity to shape changes. The Dice loss is a region-related loss that focuses more on foreground areas during training. It can effectively improve the sensitivity of the network to target areas and is suitable for conditions with a pixel category imbalance. However, the drawback of the Dice loss is that it is not sensitive to boundary characterization and focuses mainly on the interior of the mask. As a measure of shape similarity, the SD loss can complement the deficiencies of the Dice loss in order to reduce training instability. The hybrid loss function is expressed as
(1)$$Loss={Loss}_{Dice}+\alpha {Loss}_{SD}$$
(2)$${Loss}_{Dice}=1-1/C\sum_{c=1}^{C}\frac{\sum_{i}P_{i}^{c}G_{i}^{c}}{\sum_{i}{\left(P_{i}^{c}\right)}^{2}+\sum_{i}{\left(G_{i}^{c}\right)}^{2}}$$
(3)$${Loss}_{SD}=1/C\sum_{c=1}^{C}\frac{1}{V}\sum_{i}{\left(P_{i}^{c}-G_{i}^{c}\right)}^{2}{\left(D_{i}^{c}\right)}^{\gamma }$$
where $P_{i}$ and $G_{i}$ represent the predicted SoftMax probabilities and the gold standard label of voxel i in channel C, respectively; $D_{i}$ is the corresponding normalized distance to the gold standard surface; and γ and α are parameters adjusting the surface error penalty; the weight of ${Loss}_{SD}$ is set to 1.

## 3. Experiments and Results

### 3.1. Experimental Dataset

The experimental data for this study were provided by the Third Affiliated Hospital of Guangzhou Medical University and comprised 60 cardiac CT image samples. All scans were performed using a Siemens SOMATOM Force third-generation dual-source dual-energy spiral CT scanner with tube voltages ranging from 70 to 110 kV, current ranging from 12 to 85 mA, slice thicknesses of 5 mm, slice intervals of 5 mm, pitch of 1.2, and temporal resolution of 250 ms. The number of slices in each case ranged from 108 to 152, with an average of 130, and a slice size of 512 × 512 pixels. The dataset had a size of 60 × 130 × 512 × 512 (60 cases, 130 slices per case, and 512 × 512 pixels per slice), with a ratio of 6:2:2 for random partitioning into training, validation, and testing sets.

### 3.2. Image Preprocessing

The cardiac CT images were stored in Hounsfield units (HU) per pixel, with different tissue organs typically corresponding to different HU ranges. To improve the local contrast of the images, specific HU windows were first used to perform gray-level truncation of the original images. Normalization was performed by setting pixel values greater than 2048 to 2048 and those less than 0 to 0, followed by division by 2048. To reduce the risk of overfitting, the CT images were augmented. To avoid alteration of clinical features such as shape, the data augmentation methods included random horizontal flip, random horizontal and vertical translations (5%), and random scale (95–105%), with the aspect ratio preserved.

### 3.3. Functionally Gradient Porous Mandibular Prosthesis

The experimental hardware environment consisted of an Intel^®^ Xeon^®^ E5-2678 v3 CPU and an NVIDIA GTX-1080Ti GPU with 11 GB memory, running on the Ubuntu 18.04 operating system and programmed with Python 3.7. All the programs were implemented using the PyTorch 2.0.1 open-source framework. The network parameters in the experiment included an input image size of 512 × 512, a batch size of 12, and an initial learning rate set at 0.0001; the model optimizer was adaptive moment estimation (Adam: A Method for Stochastic Optimization) [34], and the maximum number of iterations was 300. During the training process, if the loss value did not decrease for ten consecutive rounds on the validation set, the learning rate was halved. To better verify the performance of the network model, five-fold cross-validation was used to obtain the average as the final experimental result.

### 3.4. Evaluation Metrics

The Dice similarity coefficient (DSC) [35] and Hausdorff distance (HD) [36], which are commonly used in segmentation tasks in medical imaging, were used as quantitative evaluation metrics for segmentation accuracy assessment. At the same time, Recall and Precision and a total of four evaluation metrics were used as quantitative evaluation indicators of the model. The formula for calculating Dice is expressed as
(4)$$Dice(X,Y)=\frac{2|G\cap P|}{\left|G\right|\cup |P|}$$
where G and P represent the manually segmented mask and the prediction mask, respectively, using binary tags. Dice calculates the ratio of twice the intersection of two masks to their union, which reflects the similarity between the target region of segmentation and the annotated target region. The higher the similarity, the better the segmentation effect. Dice ranges from 0 to 1, where 1 represents the best segmentation, and 0 represents the worst segmentation.

The HD is the maximum distance between a point in the automatic segmentation group and the closest point in the corresponding manual segmentation group and is expressed in millimeters. It was used to calculate the distance between the boundary of the actual value and that of the predicted region. It can be expressed as
(5)$${HD}_{95}=max(h\left(G,P\right),h\left(P,G\right))$$
(6)$$h\left(G,P\right)=\max min|\left|g-p\right||$$
where G and P represent the volumes of the gold standard and predicted regions, respectively; their corresponding contour points are denoted as G = {$g_{0}$, . . . . . . , $g_{n}$} and P = {$p_{0}$, . . . . . . , $p_{n}$}, respectively. The ${HD}_{95}$ ranges from zero to infinity.

Recall is also known as sensitivity; it refers to the ability of a segmentation model to accurately identify and include all the relevant regions or structures of interest in an image. It quantifies the proportion of true positive pixels or voxels that are correctly classified as belonging to the target region, out of all the pixels or voxels that actually belong to the target region. Its calculation formula is as follows, where the true segmentation result is G and the actual segmentation result is P.
(7)$$Recall=\frac{G\cap P}{G}$$

Precision is expressed as
(8)$$Precision=\frac{G\cap P}{P}$$

A higher DSC, Recall, Precision, and a lower ${HD}_{95}$ indicate more accurate segmentation. In this study, the DSC, Recall, Precision, and ${HD}_{95}$ were calculated for each case in three dimensions.

### 3.5. Experimental Results

#### 3.5.1. Impact of Grouping Method for Cardiac Substructures on Segmentation Results

We divided the ten cardiac substructures into coarse and fine segmentation groups based on the grouping methods presented in Table 1. With no prior knowledge, we tested the segmentation accuracy of the cardiac substructures for the three grouping methods using two consecutive segmentation steps. The results in Table 3 indicate that the segmentation results of the second grouping method were the best. For the four relatively large substructures, the left atrium, right atrium, left ventricle, and right ventricle, the difference in the segmentation results was not significant, and the Dice coefficients were all greater than 0.8. For the three medium-sized substructures, including the ascending aorta, descending aorta, and pulmonary artery, most of the Dice coefficients were greater than 0.8. Regarding the three smaller substructures, the PV, IVC, and SVC, there were noticeable differences in the Dice coefficients between the three groups. The Dice coefficient for the IVC in the first group was 0.662; for the other two veins, SVC and PV, it was 0, indicating no predicted results. In the third group, the Dice coefficients for both the SVC and IVC were 0; for the PV, it was 0.275. In contrast, in the second group, the Dice coefficient for the SVC exceeded 0.6; although the Dice coefficients for the IVC and PV were not high, they indicated definite prediction results, surpassing the segmentation performance of the first and third groups. From the experimental results, the segmentation effect of the second group was the best; thus, we selected the second grouping method as the strategy for our segmentation model.

#### 3.5.2. Segmentation Results

To verify the impact of our method on the segmentation performance, we compared the baseline U-Net, two-stage U-Net, and our method; the experimental results are shown in Table 4. It is observed that U-Net segmented most of the substructures in the CT images, including the four chambers of the heart, left atrium, right atrium, left ventricle, right ventricle, three arteries, ascending aorta, descending aorta, and pulmonary artery. These substructures had a high contrast, regular shape, and large size, resulting in better DSC, Recall, Precision, and HD scores. Three veins (PV, IVC, and SVC) were difficult to identify in the CT images due to their small size and low contrast. They were not segmented properly, leading to under-segmentation. The two-stage U-Net segmentation network divided the size of the structures into two groups: larger and smaller. This grouping made it easier to obtain clear segmentation results for smaller structures such as the PV, IVC, and SVC. Thus, we used the coarse segmentation results as the input to the fine segmentation network, embedded the anatomical structure of the four chambers into the next-level network, and guided and trained the small-scale substructures. The results showed that the proposed method significantly improved the segmentation accuracy of the heart substructures. The Dice scores of the three arteries (AA, DA, and PA) generally increased by more than 10%; the DSC of the IVC and SVC increased by over 80%; the IVC and SVC increased by more than 20%; and the PV increased by more than 40%, fully demonstrating the effectiveness of the proposed method for improving the accuracy of cardiac substructure segmentation.

Figure 6 shows the experimental results for some slices. Column (a) shows the expert manual segmentation results; column (b) shows the segmentation results of the baseline U-Net; column (c) shows the segmentation results of the two-stage U-Net, and column (d) shows the segmentation results obtained using the proposed method. The experimental results of our model are smooth without obvious under-segmentation or over-segmentation. The segmentation results for several substructures were closest to the ground truth, especially for the PV, IVC, and SVC, which were small but achieved a segmentation performance comparable to that of the larger substructures.

#### 3.5.3. Comparison with Other Deep Learning Methods

To further validate the performance of the proposed method, it was compared with other deep learning methods, including 3D U-Net [37] and nnU-Net [38]. Three-dimensional U-Net replaces the two-dimensional operations in ordinary U-Net with three-dimensional operations, such as 3D convolution and 3D pooling, which can be conveniently applied to three-dimensional data. nnU-Net, an emerging fully automatic segmentation model in medical imaging in recent years, has achieved a fully automated process for data processing and model training. The cascaded 3D U-Net has achieved positive results in medical image segmentation and is an outstanding U-Net variant model.

Table 5 presents the segmentation results for the ten cardiac substructures in the CT dataset for 3D U-net, nnU-Net, and the proposed method. In traditional cardiac segmentation, such as in the atria and ventricles, our method showed a slight improvement in the RA, LV, LA, and RV. Our method demonstrated a significant advantage in the segmentation of PA, DA, and AA. In contrast, for the three classes with few samples and small targets, SVC, IVC, and PV, our segmentation performance had a more obvious advantage. The Dice scores of the 3D U-Net and nnU-Net segmentation were not larger than 0.6, indicating that they did not achieve adequate segmentation for these three targets. The analysis shows that our method has a higher segmentation accuracy than the other methods and achieves the best segmentation effect.

## 4. Discussion

In this study, we developed a cardiac substructure segmentation framework that combines anatomical structure priors to accurately segment ten cardiac substructures including the LA, RA, LV, RV, SVC, IVC, PA, PV, AA, and DA. The framework comprises a two-step cascade network from coarse to fine segmentation, incorporating coarse segmentation results as prior information and combining them with the original image to form a multichannel input. Thus, the anatomical knowledge of large-scale structures is embedded into a fine segmentation network to guide and train small-scale structures. Compared to existing segmentation models, our framework demonstrates superior segmentation performance, particularly in multi-target segmentation tasks involving small targets.

We found that the segmentation targets had significant differences in size (for example, the largest substructure (LV) was ten times larger than the smallest substructure (SVC)), unbalanced sample distribution, different grayscales, fuzzy boundaries, irregular shapes, and non-fixed substructure positions. It is difficult to accurately identify small substructures using current end-to-end segmentation neural network models. To address this issue, we proposed a framework that differentiates substructures with substantial size discrepancies and groups them for processing to mitigate the imbalance problem. If we treat large and small substructures equally and rely solely on high-performance hardware platforms and a large amount of labeled data to train the models, we may encounter lower segmentation accuracy for small substructures with low contrast. It is difficult to obtain large amounts of matched labeled data due to availability and the cost of medical imaging data and manual labeling. Thus, we divided the ten substructures into two groups: a coarse segmentation group for easy-to-segment substructures and a fine segmentation group for difficult-to-segment substructures. We explored the inherently stable relationships between heart substructures through experiments and selected the best grouping strategy for optimal model performance. As shown in Table 2, using the left atrium, right atrium, left ventricle, and right ventricle as the coarse segmentation group and the superior vena cava, inferior vena cava, pulmonary artery, pulmonary vein, ascending aorta, and descending aorta as the fine segmentation group was optimal. This is mainly because the relative positions of the four larger structures were fixed and their size differences were not significant, making them easier to segment. The fine segmentation group consisted of arteries and veins that were connected to each other, had a tight relationship, and did not differ significantly in size. For the three veins, the smallest substructures with the lowest contrast and the most challenging structures to segment, grouping large and small substructures separately can optimize the organ segmentation imbalance and simplify tasks, making it easier to train network models and significantly improve the segmentation results.

Hierarchical segmentation enhances performance; however, it remains insufficient for tiny substructures like the superior vena cava, inferior vena cava, and pulmonary veins. Their distribution is concentrated, and coarse segmentation results encompass these small substructures within the targeted area. This inspired us to use the large substructures surrounding small substructures as prior information in small substructure segmentation to further improve segmentation accuracy. We used the first-stage segmentation results and original images as inputs to form a multichannel input for the fine segmentation network, embedding prior knowledge of the position, shape, and scale of the large substructures into the fine segmentation network. These high-resolution features were used to refine the segmentation results. Our prior knowledge included the masks of the four larger substructures, atria, and ventricles, which contained the shape and positional information of the larger structures that served as important prior knowledge for training the smaller substructures through geometric constraints. However, our prior knowledge was obtained through deep learning without additional modules or manually designed or preconceived prior information, resulting in no significant computational burden or increased computation time. By introducing the necessary prior information and increasing the number of input channels, the performance of the model was strengthened with limited data without reducing the operating efficiency of the segmentation network. The results in Table 4 indicate that the segmentation framework incorporating anatomical priors can improve the sensitivity of the network to small target regions, handle erroneous segmentation caused by image noise, and increase the reliability of the algorithm.

We combined anatomical structure priors in our cardiac substructure segmentation framework, which is compatible with base networks such as ResNet, DenseNet, and U-Net, which are commonly used for multi-target segmentation. We chose U-Net as the backbone owing to its simple and symmetrical structure, suitability for small-sample data, and excellent segmentation performance combining multiple scales of features. To further verify the effectiveness of our approach, we compared it with other deep learning methods, including U-Net, 3D U-Net, and nnU-Net, which are currently the most widely used and best-performing deep learning networks in the field of medical image segmentation.

From Table 5, it is observed that although U-Net, 3D U-Net, and nnU-Net can roughly segment most of the substructures in the CT images, it is difficult to obtain accurate segmentation boundaries for the three small veins due to significant differences in grayscale, shape, and size features across different slices, making them prone to under-segmentation or poor prediction. Compared to the other three networks, our proposed method achieved results closer to those of expert manual segmentation. From the Dice evaluation metric, the segmentation accuracy of our method was significantly better than that of the other methods, particularly for the SVC, IVC, and PV substructures, which had limited samples and were small targets. There were two reasons: first, we divided the ten substructures into two groups based on size and difficulty, a coarse segmentation group (including larger substructures such as the four chambers) and a fine segmentation group (including smaller substructures such as arteries and veins). In the first stage, segmentation was performed on the coarse segmentation group; these four targets were relatively large with fixed relative positions, with better segmentation results. The first-stage segmentation results and the original image were used as inputs, forming a multichannel input for the fine segmentation network. First, prior knowledge of the location, shape, and scale of the larger substructures was embedded in the fine segmentation network, and high-resolution feature maps were used to refine the segmentation results. By cascading these two segmentation tasks, each level of the network model was easier to train. Using the segmentation results of the previous network to constrain the segmentation of the subsequent network can effectively reduce the number of false positives outside the boundary. Thus, the proposed coarse-to-fine cascaded segmentation framework is theoretically and practically reliable.

This study has several limitations. With the high cost of medical image data labeling, the size of the dataset was limited. The generalization ability of the model was not tested because the experiment lacked external test datasets; there may be differences in modeling accuracy owing to different CT scanning parameters. In future studies, we will continue to improve our method in order to address these issues. More data will be collected to enhance the generalization ability of the method. The segmentation network was adaptively adjusted according to specific characteristics of the cardiac substructures to further improve the overall segmentation accuracy and meet clinical application requirements.

## 5. Conclusions

This study proposes a cardiac substructure segmentation framework that integrates prior anatomical structure knowledge to address challenges such as varying grayscales, blurred boundaries, irregular shapes, and inconsistent positions of key substructures in cardiac CT images. The proposed method employs a cascading approach to automatically segment ten substructures in cardiac CT images, achieving accurate segmentation results for targets with different scales, from coarse to fine. The presented method leverages prior knowledge of the heart substructures’ location, shape, and size to separately process differently scaled structures. It employs a hierarchical segmentation approach to enhance segmentation performance, particularly for smaller structures. This segmentation model fuses the benefits of coarse-to-fine strategies and U-Net. By employing a cascading approach to progressively segregate larger to smaller substructures in cardiac CT images, the method effectively mitigates the issue of negligibly small substructures. Moreover, based on the coarse segmentation results, prior knowledge of the positions, shapes, and sizes of larger substructures was embedded into fine segmentation networks to refine the segmentation of smaller substructures. The proposed method eliminates the need for manual intervention and accurately segments substructures of varying sizes, shapes, and positions in cardiac CT images, even in cases of low contrast and blurred boundaries. The experiments demonstrated that this method can effectively identify small targets within substructures, outperforming existing methods in terms of segmentation accuracy.

## Figures and Tables

**Figure 1 bioengineering-10-01267-f001:**
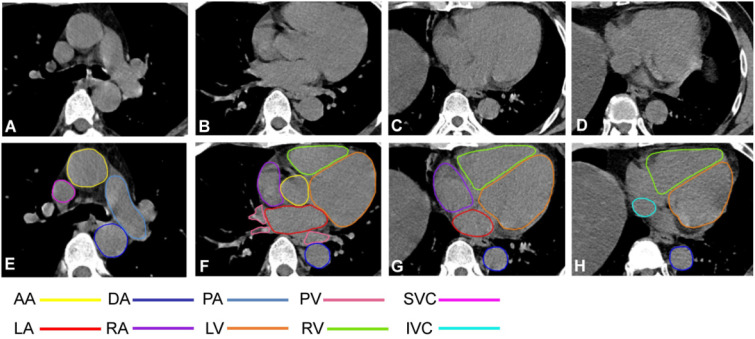
Contour labeling of cardiac substructure. (**A**–**D**) Plain chest CT (slice thickness: 5 mm, mediastinal window); (**E**,**F**) cardiac substructure segmentation based on chest CT scan; (**A**,**E**) level under tracheal carina; (**B**,**F**) ascending aortic root level; (**C**,**G**) four-cavity heart level; (**D**,**H**) coronary sinus level.

**Figure 2 bioengineering-10-01267-f002:**
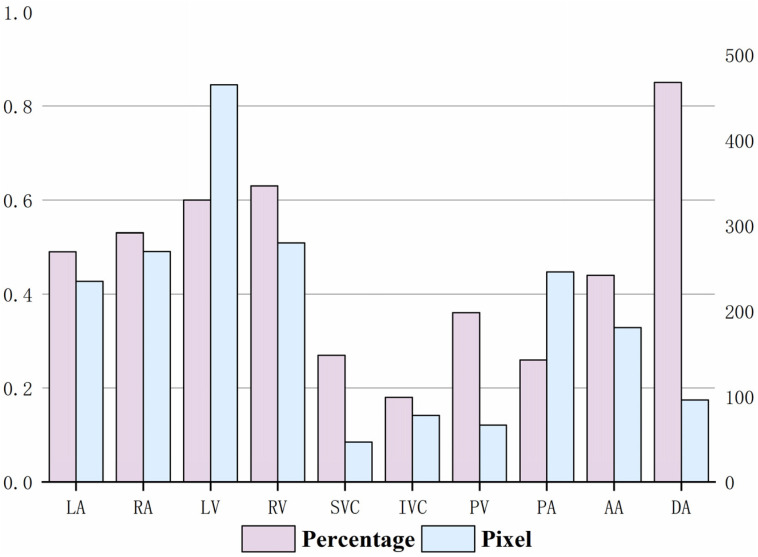
Proportion of each substructure and average size in dataset.

**Figure 3 bioengineering-10-01267-f003:**
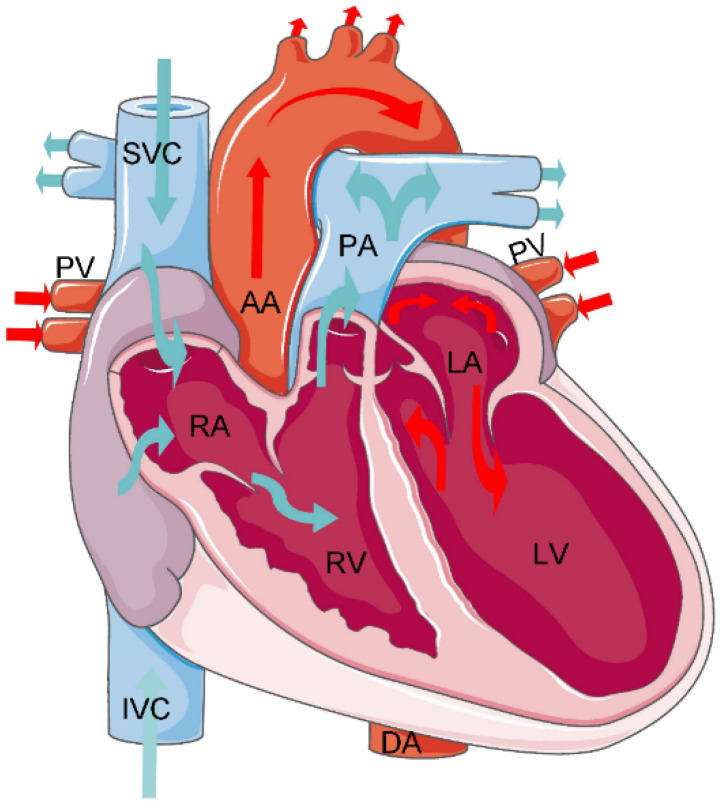
Cardiac anatomy.

**Figure 4 bioengineering-10-01267-f004:**
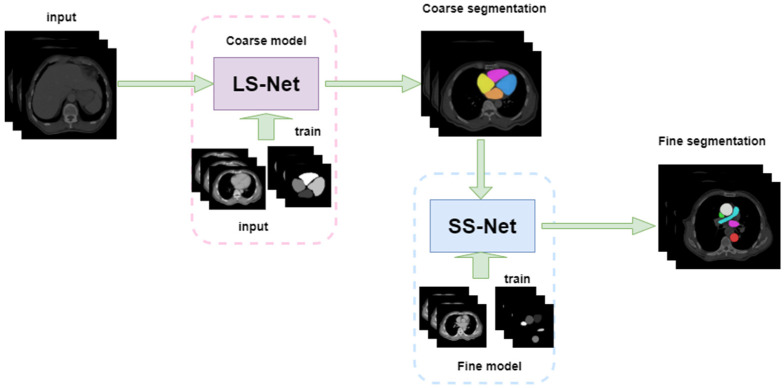
Cardiac substructure segmentation framework.

**Figure 5 bioengineering-10-01267-f005:**
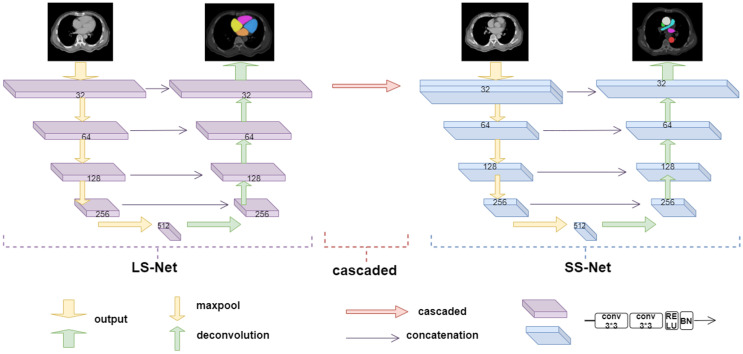
Architecture of proposed LS-Net and SS-Net.

**Figure 6 bioengineering-10-01267-f006:**
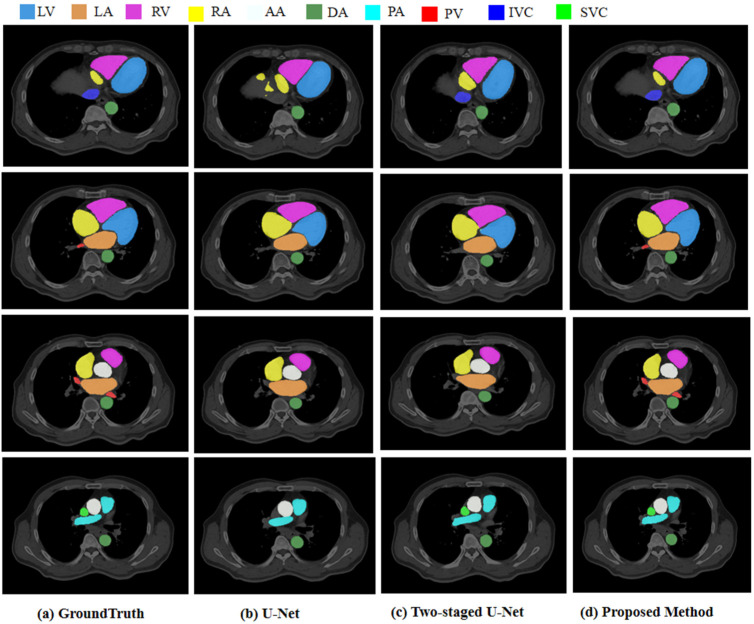
Segmentation comparisons for U-Net, two-staged U-Net, and the proposed method.

**Table 1 bioengineering-10-01267-t001:** The performance parameters for porous scaffolds, covering porosity and peak stress.

Substructures	Acronym	Anatomical Position
Left Ventricle	LV	The left ventricle is located in the lower left part of the heart, below the left atrium, and in the left rear of the right ventricle, which is conical. In the four-chamber view of transverse CT, the left ventricle is located in the upper left.
Right Ventricle	RV	The right ventricle is located in the lower right part of the heart, in the anterior lower part of the right atrium. In the four-chamber view of transverse CT, the right ventricle is located in the upper right.
Left Atrium	LA	The left atrium is located in the upper left part of the heart and is the most posterior heart cavity. In the four-chamber view of transverse CT, the left atrium is located in the lower left.
Right Atrium	RA	The right atrium is located in the upper right part of the heart, on the right and anterior side of the left atrium. In the four-chamber view of transverse CT, the right atrium is located in the lower right.
Ascending Aorta	AA	The ascending aorta is connected to the left ventricle. In CT, the aorta is the largest cardiac blood vessel in the mediastinum and is presented in a circular structure in the transverse position.
Descending Aorta	DA	In CT, the descending aorta is located beside the spine and has a circular structure.
Pulmonary Artery	PA	The pulmonary artery starts from the bottom of the right ventricle. In the transverse CT, the left and right pulmonary arteries extend from the main pulmonary artery to both sides, presenting a tree structure.
Pulmonary Vein	PV	On the axial CT, the pulmonary veins extend to both sides, showing a ‘reptile‘ shape due to the small diameter.
Inferior Vena Cava	IVC	In the four-chamber view of transverse CT, the inferior vena cava is often located at the bottom of the heart and is closely related to the liver.
Superior Vena Cava	SVC	The superior vena cava is located on the right side of the ascending aorta in the transverse CT; its diameter is smaller than that of the ascending aorta.

**Table 2 bioengineering-10-01267-t002:** Grouping methods of cardiac substructures.

Group	Coarse Segment Group	Fine Segment Group
1	LV RA AA DA SVC IVC	LA RV PV PA
2	LA LV RA RV	AA DA PA PV SVC IVC
3	LA LV PV AA DA	RA RV IVC SVC PA

**Table 3 bioengineering-10-01267-t003:** Segmentation results for different grouping methods of cardiac substructures. Bold indicates the best result.

Grouping	LA	RA	LV	RV	SVC	IVC	PA	PV	AA	DA
Group 1	**0.884**	0.846	0.806	0.851	0.0003	**0.662**	**0.843**	0.0003	**0.8** **92**	**0.8** **75**
Group 2	0.853	**0.851**	**0.879**	**0.868**	**0.6** **41**	0.520	0.808	**0.** **362**	0.856	0.862
Group 3	0.823	0.837	0.818	**0.832**	0.0002	0.0005	0.788	0.275	0.871	0.870

**Table 4 bioengineering-10-01267-t004:** DSC and HD95 comparisons for U-Net, two-staged U-Net, and the proposed method on all ten substructures in testing data. Bold indicates the best result or significant differences.

Substructures	DSC			Recall			Precision			HD/mm		
	U-Net	Two -Staged	Proposed Method	U-Net	Two -Staged	Proposed Method	U-Net	Two -Staged	Proposed Method	U-Net	Two -Staged	Proposed Method
LA	0.809	0.853	**0.853**	0.779	0.840	**0.840**	0.847	0.850	**0.850**	0.355	0.301	**0.301**
RA	0.837	0.851	**0.851**	0.842	0.875	**0.875**	0.834	0.837	**0.837**	0.492	0.422	**0.422**
LV	0.878	0.879	**0.879**	0.870	0.871	**0.871**	0.887	0.889	**0.889**	0.352	0.367	**0.367**
RV	0.856	0.868	**0.868**	0.825	0.873	**0.873**	0.856	0.864	**0.864**	0.411	0.383	**0.383**
SVC	0.355	0.641	**0.816**	0.574	0.650	**0.801**	0.250	0.676	**0.832**	3.874	1.887	**0.345**
IVC	0.302	0.520	**0.801**	0.208	0.459	**0.818**	0.551	0.711	**0.876**	3.912	2.603	**0.366**
PA	0.804	0.808	**0.902**	0.764	0.767	**0.909**	0.854	0.864	**0.895**	0.548	0.554	**0.284**
PV	0.190	0.362	**0.797**	0.258	0.379	**0.754**	0.422	0.577	**0.835**	5.237	3.431	**0.351**
AA	0.850	0.856	**0.928**	0.823	0.817	**0.903**	0.884	0.930	**0.954**	0.479	0.392	**0.213**
DA	0.856	0.862	**0.873**	0.826	0.829	**0.936**	0.909	0.912	**0.818**	0.411	0.346	**0.312**

**Table 5 bioengineering-10-01267-t005:** Comparisons of 3D U-net, nnU-Net, and the proposed method. Bold indicates the best result.

Substructures	DSC			Recall			Precision			HD/mm		
	3D U-Net	nnU-Net	Proposed Method	3D U-Net	nnU-Net	Proposed Method	3D U-Net	nnU-Net	Proposed Method	3D U-Net	nnU-Net	Proposed Method
LA	0.833	0.842	**0.853**	0.815	0.821	**0.840**	0.835	0.844	**0.850**	0.473	0.315	**0.301**
RA	0.820	0.817	**0.851**	0.821	0.832	**0.875**	0.817	0.827	**0.837**	0.513	0.521	**0.422**
LV	0.871	**0.882**	0.879	0.863	**0.878**	0.871	0.873	**0.890**	0.889	0.357	0.372	**0.367**
RV	0.792	0.831	**0.868**	0.791	0.829	**0.873**	0.784	0.829	**0.864**	0.589	0.551	**0.383**
SVC	0.712	0.780	**0.816**	0.722	0.750	**0.970**	0.723	0.776	**0.832**	0.602	0.574	**0.345**
IVC	0.568	0.782	**0.801**	0.584	0.774	**0.801**	0.591	0.791	**0.876**	2.801	0.574	**0.366**
PA	0.751	0.792	**0.902**	0.742	0.757	**0.818**	0.766	0.784	**0.895**	0.586	0.554	**0.284**
PV	0.527	0.546	**0.797**	0.526	0.535	**0.909**	0.521	0.577	**0.835**	3.152	3.431	**0.351**
AA	0.858	0.876	**0.928**	0.846	0.864	**0.754**	0.860	0.910	**0.954**	0.447	0.392	**0.213**
DA	0.802	0.839	**0.873**	0.809	0.821	**0.903**	0.813	0.827	**0.818**	0.351	0.346	**0.312**

## Data Availability

Not applicable.

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
