# Peer review of "Anatomical Prior-Based Automatic Segmentation for Cardiac Substructures from Computed Tomography Images"

_bioengineering, 2023, doi:10.3390/bioengineering10111267_

Round 1

Reviewer 1 Report

Comments and Suggestions for Authors

I have the following concerns.

1. In addition to the Hausdorff distance, there are other measures of difference, such as Shannon entropy, Kullback-Leibler divergence. Justify your choice.

2. The comparisons given in Table 4 based only on the DSC and HD metrics are insufficient. Show the comparison of the effectiveness of different architectures and models in ML based on other metrics, for example, precision, recall, F1 score.

3. As a rule, semantic segmentation is used more and more recently in the processing of medical images. Why didn't you use it.

4. Cascade algorithms have long been used for image classification in ML. What is your difference.

5. This distribution between training, validation and testing samples 7:2:1 does not correspond to the classic approach in ML. The ratio should be the following 6:2:2.

6. There are no comparisons on the runtime parameter.

7. References must be supplemented with articles for 2021-2023.

Comments on the Quality of English Language

 Minor editing of English language required

Reviewer 2 Report

Comments and Suggestions for Authors

The herein presented paper/study deals with a new and interesting topic.

The results presented are quite nice even though they are preliminary and not directly usable in clinical practice.

Comments on the Quality of English Language

Just minor review is necessary.

Author Response

Thank you very much for the time and effort you put into our manuscript. 

Reviewer 3 Report

Comments and Suggestions for Authors

These are some comments regarding the methodology included in the proposed manuscript.

The patient’s database is extremely small for the training and the evaluation processes of such a multiple and highly adjustable methodology.

The anatomical description f the heart is a basic knowledge and is quite extensive –covers a large area of the manuscript

It is not obvious which patient’s dataset has been utilized for the evaluation process. It should be clear that in the evaluation-validation process should utilized data that haven’t been integrated previously in any other process. Otherwise, the outcome would be biased.

The differentiation of the training and the testing dataset should be checked especially throughout the cascade segmentation framework.

The difficulty of the technique in the identification of small veins should be reconsidered since it introduces questions about the overall performance of the method.

It seems that the whole method locks of generalization ability since its performance depends greatly on patient’s data, image (CT) quality and algorithmic and training parameters that promotes over fitting behavior.       

Round 2

Reviewer 1 Report

Comments and Suggestions for Authors

I am satisfied with the responses to my comments. The changes and additions made have greatly improved the perception of the results obtained.

Comments on the Quality of English Language

 Minor editing of English language required

Author Response

(The authors gave the same response as above.)

Reviewer 3 Report

Comments and Suggestions for Authors

I’d like to acknowledge the authors for the feedback given to the initial questions. Most of the critical questions have been answered, justifying the difficulties of the specific study. However, most of the limitations that arose during the present research still remains based mainly on the evaluation and the accuracy of the overall technique. Since the present study constitutes a primary approximation of the specific methodology, it could be able to provide the basis for its further development and improvements.

Author Response

(The authors gave the same response as above.)
